# Global sensitivity analysis of simulated polarimetric remote sensing observations over snow

**Matteo Ottaviani**[1,2]**, Gabriel Harris Myers**[3]**, and Nan Chen**[4]

[1]NASA Goddard Institute for Space Studies, New York, NY 10025, USA
[2]Terra Research Inc, Hoboken, NJ 07030, USA
[3]Courant Institute of Mathematical Sciences, New York University, New York, NY 10012, USA
[4]Stevens Institute of Technology, Hoboken, NJ 07030, USA

**Correspondence:** Matteo Ottaviani (matteo.ottaviani@nasa.gov)

**Abstract.** This study presents a detailed theoretical assessment of the information content of passive polarimetric observations over snow scenes, using a global sensitivity analysis (GSA) method. Conventional sensitivity studies focus on varying a single parameter while keeping all other parameters fixed. In contrast, the GSA correctly addresses the covariance of state parameters across their entire parameter space, hence favoring a more correct interpretation of inversion algorithms and the optimal design of their state vectors.

The forward simulations exploit a vector radiative transfer model to obtain the Stokes vector emerging at the top of the atmosphere for different solar zenith angles, when the bottom boundary consists of a vertically resolved snowpack of nonspherical grains. The presence of light-absorbing particulates (LAPs), either embedded in the snow or aloft in the atmosphere above in the form of aerosols, is also considered. The results are presented for a set of wavelengths spanning the visible (VIS), near-infrared (NIR), and shortwave infrared (SWIR) region of the spectrum.

The GSA correctly captures the expected, high sensitivity of the reflectance to LAPs in the VIS–NIR and to grain size at different depths in the snowpack in the NIR–SWIR. With adequate viewing geometries, mono-angle measurements of total reflectance in the VIS–SWIR (akin to those of the Moderate Resolution Imaging Spectroradiometer, MODIS) resolve grain size in the top layer of the snowpack sufficiently well. The addition of multi-angle polarimetric observations in the VIS–NIR provides information on grain shape and microscale roughness. The simultaneous sensitivity in the VIS–NIR to both aerosols and snow-embedded impurities can be disentangled by extending the spectral range to the SWIR, which contains information on aerosol optical depth while remaining essentially unaffected when the same particulates are mixed with the snow. Multi-angle polarimetric observations can therefore (i) effectively partition LAPs between the atmosphere and the surface, which represents a notorious challenge for snow remote sensing based on measurements of total reflectance only and (ii) lead to better estimates of grain shape and ice crystal roughness and, in turn, the asymmetry parameter, which is critical for the determination of albedo. The retrieval uncertainties are minimized when the degree of linear polarization is used in place of the polarized reflectance.

The Sobol indices, which are the main metric for the GSA, were used to inform the choice of state parameters in retrievals performed on data simulated for multiple instrument configurations. Improvements in retrieval quality with the addition of measurements of polarization, multi-angle views, and different spectral channels reflect the information content, identified by the Sobol indices, relative to each configuration.

The results encourage the development of new remote sensing algorithms that fully leverage multi-angle and polarimetric capabilities of modern remote sensors. They can also aid flight planning activities, since the optimal exploitation of the information content of multi-angle measurements depends on the viewing geometry. The better characterization of surface and atmospheric parameters in snow-covered regions advances research opportunities for scientists of the

cryosphere and ultimately benefits albedo estimates in climate models.

## 1 Introduction

The quantification of the surface energy balance of snow-covered regions is of extreme importance for Earth-system model simulations aimed at global climate studies (Hansen and Nazarenko, 2004; Fettweis et al., 2008; van den Broeke et al., 2011; Rae et al., 2012; van Angelen et al., 2012; Tedesco et al., 2013; Alexander et al., 2014; Colgan et al., 2014). Since snow albedo fundamentally depends on the optical and microphysical properties of ice crystals (Wiscombe and Warren, 1980; Aoki et al., 2000; Flanner and Zender, 2006; Bougamont et al., 2007; Dang et al., 2016; He et al., 2018) and light-absorbing impurities potentially present in the snowpack (Warren and Wiscombe, 1980; Hansen and Nazarenko, 2004; Dumont et al., 2014), better knowledge of the properties of such components and their evolution is a high-priority objective for the modeling of the cryosphere (Tedesco et al., 2013; Dumont et al., 2014).

One fundamental source of uncertainty in the remote sensing of these properties is the treatment of snow as a collection of spherical grains (Tanikawa et al., 2020). Although useful in some contexts, such approximation can underestimate the albedo by a few percent (Xie et al., 2006; Tedesco and Kokhanovsky, 2007; Libois et al., 2013; Tedesco et al., 2013; Dumont et al., 2014; Dang et al., 2016; Räisänen et al., 2017), a discrepancy that can be exaggerated by snow albedo feedback processes (Thackeray et al., 2018; Colman, 2013; Hansen and Nazarenko, 2004). To avoid this assumption we employ hexagonal prisms, which have been demonstrated (van Diedenhoven et al., 2012) to serve well as radiative proxies for more complex shapes, while having the advantage of being characterized only by their aspect ratio (AR; with AR > 1 for columns and AR < 1 for plates) and the microscale roughness ($D$) of the crystal facets. The implementation of this methodology in advanced radiative transfer (RT) models has produced successful retrievals for parameters that describe the crystals forming ice clouds (van Diedenhoven et al., 2014b) and the reflectance properties of snow-covered surfaces (Ottaviani et al., 2012, 2015) from data collected with the NASA Goddard Institute for Space Studies (GISS) airborne Research Scanning Polarimeter (RSP; Cairns et al., 1999).

Another major challenge is the determination of light-absorbing particulate (LAP) content and its partitioning between LAPs deposited in snow versus those suspended above in the form of atmospheric aerosols (Warren, 2013). Because the polarization state of light is also sensitive to this partitioning (Ottaviani, 2022), remote sensors like the RSP; the Second-generation Global Imager (SGLI; Tanaka et al., 2018); the polarimeters launched aboard the Plankton,

Aerosol, Cloud, ocean Ecosystem (PACE) mission (Werdell et al., 2019; Hasekamp et al., 2018); and the upcoming Multi-viewing, Multi-channel, Multi-polarization imaging mission (3MI) (Biron et al., 2013; Marbach et al., 2013) offer augmented retrieval capabilities.

Zhang et al. (2023) have recently evaluated the performance of a snow kernel introduced into an inverse algorithm in retrieving the microphysics of aerosols above snow, based on observations of the Polarization and Directionality of the Earth's Reflectances (POLDER) spaceborne sensor, which flew from 2004 to 2013. However, their study does not address the retrieval of the microphysical properties of the snowpack and the distribution of LAPs between the snow and the atmosphere or information content aspects. This paper extends the studies presented in Ottaviani (2022), examining these details via a global sensitivity analysis (GSA) of simulated top-of-the-atmosphere (TOA) polarimetric observations.

Section 2 explains the setup of the RT calculations needed to generate the look-up table (LUT) of the Stokes vectors at the TOA produced for random combinations of the input parameters and outlines the adaptation of the GSA formalism to the scopes of this analysis. The results of the GSA are presented in Sect. 3 along with inverse retrievals of spaceborne observations simulated for different observational configurations. We first consider pure-snow scenes to highlight the sensitivity to the ice crystal properties. We then address more realistic remote sensing scenarios where the atmosphere is allowed to contain a layer of absorbing aerosols, and the snowpack contains impurities. The paper concludes with some recommendations for operational retrievals.

## 2 Methods

### 2.1 Radiative transfer simulations

The plane-parallel RT code employed to generate the LUT is based on the general doubling–adding formalism described by De Haan et al. (1987). It features consistent treatment of the radiative effects deriving from atmospheric molecular scattering, aerosols and clouds, and any surface whose reflectance is known in its analytical form or in terms of its bidirectional reflectance distribution function (BRDF) properties and the polarization counterpart (BPDF). The code has been used for decades to model measurements from the RSP over a variety of Earth scenes, including those containing ice crystals in clouds (van Diedenhoven et al., 2013) and ground snow (Ottaviani et al., 2012, 2015).

Recognizing the similarities to the polarimetric signatures of ice crystals in cirrus clouds (Ottaviani et al., 2012, 2015), the snowpack is modeled as an optically semi-infinite collection of non-spherical ice crystals at the bottom of the atmosphere so that the reflectance of the actual underlying surface is irrelevant. As with other media, the photon penetration

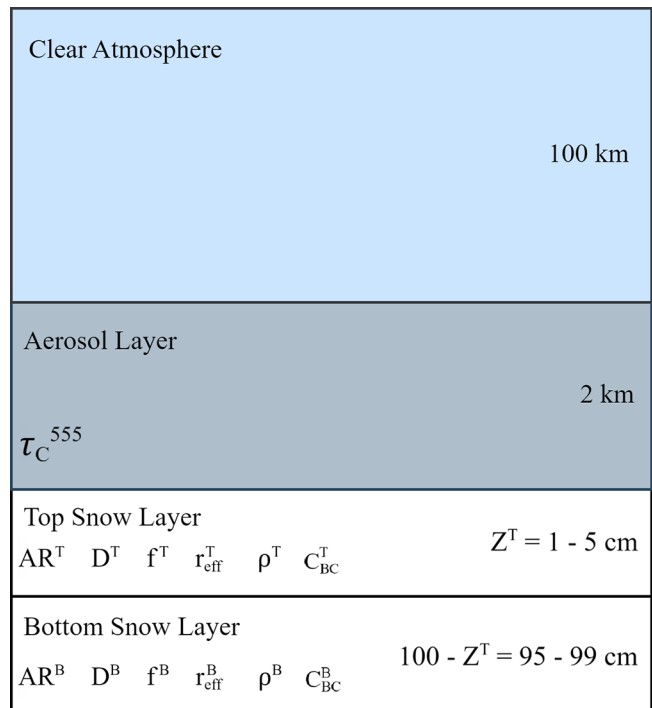

**Figure 1.** Model for the snow–atmosphere system. The snowpack is described as a thin layer sitting on an optically semi-infinite layer, for a total thickness of 1 m. The parameters in each layer are varied independently. The atmosphere is allowed to contain light-absorbing aerosols ($\tau_C^{555}$) in the lowest layer (first 2 km above the surface). See Table 1 for the complete list of model parameters and their ranges of variability.

depth in snow depends on wavelength (Kokhanovsky, 2022; Libois et al., 2013), so different instrumental channels effectively probe different depths, and this fact must be taken into account during multi-spectral retrievals (Li et al., 2001). In contrast to retrievals of grain size for mono-layer snowpacks (Nolin and Dozier, 1993; Painter et al., 2003), such an approach has been exploited to retrieve grain size in both a thin surface layer and a thick layer below using measurements from the Moderate Resolution Imaging Spectroradiometer (MODIS; Aoki et al., 2007; Painter et al., 2009). For this reason, the snowpack in our model is vertically resolved in a thick bottom layer capped by a thin top layer (see Fig. 1).

To span a wide spectral range as required by remote sensing applications, we consider channels at 411, 469, 555, 670, 864, 1589, and 2266 nm, which in the case of the RSP sensor are all equipped with polarization capabilities. Except for minimal differences in the precise centerband values, the channels in this set are also available from heritage instruments, such as the Moderate Resolution Imaging Spectroradiometer (MODIS), to favor atmospheric correction both over land and over the ocean. However, in many cases these sensors provide only total reflectance and at a single view per pixel. We include the MODIS band at 2112 nm in the list of

channels despite it being very close to 2266 nm because these wavelengths lie on the shoulder of a major absorption band where radiative differences can arise very quickly. Moreover, this channel will be available from 3MI with polarization capabilities.

Absorbing aerosols are climatologically relevant because deposition events can cause large variations in albedo (Dumont et al., 2014; Hansen and Nazarenko, 2004; Khan et al., 2023; Warren and Wiscombe, 1980). To examine the capability of different observational configurations (see Sect. 3) to distinguish them from impurities in the snow, both the lowest atmospheric layer (located within the first 2 km above the snowpack) and the snow are allowed to contain variable amounts of the same spherical LAP with properties typical of soot ($n = 1.80-0.6i$, $r_{\mathrm{eff}} = 0.11\,\mu\mathrm{m}$, $v_{\mathrm{eff}} = 0.38$) (Dubovik et al., 2002).

Finally, the presence of exclusively absorbing gases ($H_2O$, $O_3$, etc.) in the background atmosphere is neglected because it does not affect the conclusions drawn from the sensitivity study presented below. The complete list of the descriptive parameters and their bounds in the LUT can be found in Table 1.

The optical properties of the hexagonal prisms are produced via a geometric optics (GO) code (Macke et al., 1996; van Diedenhoven et al., 2012) as a function of the aspect ratio (AR), microscale roughness ($D$), and effective radius ($r_{\mathrm{eff}}$) and are integrated over a power-law size distribution (Geogdzhayev and van Diedenhoven, 2016). The microscale roughness represents the standard deviation of the distribution of angles used to randomly perturb the orientation of the ice crystal facet encountered by the incident beam in the GO calculations (van Diedenhoven et al., 2014a). Previous attempts to fit the surface contribution to the signal measured by airborne polarimeters have shown that $D \gtrsim 0.25$ (Ottaviani, 2012, 2015). Such roughness is sufficient to extinguish the halo peaks characteristic of more pristine crystals (van Diedenhoven et al., 2012).

The asymmetry parameters of columns and plates of reciprocal ARs are very similar (Ottaviani et al., 2015; van Diedenhoven et al., 2014a). To enable retrievals of representative crystal shapes and test the sensitivity to the mixing proportion, we therefore assume that the population of grains is composed of a fraction $f$ of plates (superscript "P") with aspect ratio $AR^P$ and a fraction $(1 - f)$ of columns (superscript "C") with aspect ratio $AR^C = 1 / AR^P$. The fraction is allowed to vary independently in each layer. For ice crystals of a given AR, the extinction and scattering efficiencies are

$$Q_{\mathrm{ext}} = \frac{C_{\mathrm{ext}}}{A}, \tag{1}$$

$$Q_{\mathrm{sca}} = \frac{C_{\mathrm{sca}}}{A}, \tag{2}$$

where $C_{\mathrm{ext}}$ and $C_{\mathrm{sca}}$ are the extinction and scattering coefficient, and $A$ is the projected area of the hexagonal prism (van Diedenhoven et al., 2014a). The extinction and scattering ef-

**Table 1.** Model parameters and their lower and upper bounds. The parameters describing the top and bottom snow layer are denoted with superscripts T and B and are varied independently. Soot-like LAPs are characterized by $n = 1.80 - 0.6i$, $r_{\text{eff}} = 0.11\,\mu\text{m}$, and $v_{\text{eff}} = 0.38$ (Dubovik et al., 2002; Torres et al., 2017).

| Parameter | Symbol | Bounds |
|---|---|---|
| Aspect ratio | $\text{AR}^{\text{T}}$, $\text{AR}^{\text{B}}$ | 0.037–1.0 |
| Microscale roughness | $D^{\text{T}}$, $D^{\text{B}}$ | 0.2–0.7 |
| Snow grain mixing proportion (area fraction of columns) | $f^{\text{T}}$, $f^{\text{B}}$ | 0.0–1.0 |
| Snow grain effective radius | $r_{\text{eff}}^{\text{T}}$, $r_{\text{eff}}^{\text{B}}$ | 56–2560 μm |
| Top layer snow density | $\rho^{\text{T}}$ | 0.07–0.4 g cm$^{-3}$ |
| Bottom layer snow density | $\rho^{\text{B}}$ | 0.25–0.5 g cm$^{-3}$ |
| LAP concentration in snow | $C_{\text{BC}}^{\text{T}}$, $C_{\text{BC}}^{\text{B}}$ | 0–1 ppmw |
| Aerosol optical depth (555 nm) | $\tau_{\text{C}}^{555}$ | 0.0–0.4 |
| Top layer thickness | $Z^{\text{T}}$ | 0.01–0.05 m |

ficiencies of the mixture are

$$Q_{\text{ext}}^{\text{mix}} = f \cdot Q_{\text{ext}}^{\text{C}} + (1 - f) \cdot Q_{\text{ext}}^{\text{P}}, \tag{3}$$

$$Q_{\text{sca}}^{\text{mix}} = f \cdot Q_{\text{sca}}^{\text{C}} + (1 - f) \cdot Q_{\text{sca}}^{\text{P}}, \tag{4}$$

and the corresponding phase function and asymmetry parameter are

$$P^{\text{mix}} = \left( f \cdot P^{\text{C}} \cdot Q_{\text{sca}}^{\text{C}} + (1 - f) \cdot P^{\text{P}} \cdot Q_{\text{sca}}^{\text{P}} \right) / Q_{\text{sca}}^{\text{mix}}, \tag{5}$$

$$g^{\text{mix}} = \left( f \cdot g^{\text{C}} \cdot Q_{\text{sca}}^{\text{C}} + (1 - f) \cdot g^{\text{P}} \cdot Q_{\text{sca}}^{\text{P}} \right) / Q_{\text{sca}}^{\text{mix}}. \tag{6}$$

The projected area of the mixture is assumed to be that of the column crystals so that the extinction cross section, the scattering cross section, and the single-scattering albedo of the mixture are

$$C_{\text{ext}}^{\text{mix}} = Q_{\text{ext}}^{\text{mix}} \cdot A^{\text{C}}, \tag{7}$$

$$C_{\text{sca}}^{\text{mix}} = Q_{\text{sca}}^{\text{mix}} \cdot A^{\text{C}}, \tag{8}$$

$$\text{SSA}^{\text{mix}} = Q_{\text{sca}}^{\text{mix}} / Q_{\text{ext}}^{\text{mix}}. \tag{9}$$

The impurities in the snowpack are externally mixed with the snow grains (Tanikawa et al., 2020). Their optical properties are calculated by Mie calculations inherent to the RT code, as is done for the aerosols, and are assumed to follow lognormal aerosol size distributions (Hansen and Travis, 1974). Using the layer-resolved inherent optical properties above, together with the optical depths, the RT code simulates the TOA reflectances ($R_I$, $R_Q$, $R_U$) corresponding to the first three parameters ($I$, $Q$, $U$) of the Stokes vector, describing the linear state of light polarization. Circular polarization is represented by the fourth element ($V$), which generally has negligible relevance to remote sensing applications (Kawata, 1978), so it is omitted from the analysis. The modeled reflectances can be output for any viewing geometry. We choose observations along the principal plane for which the collection of scattering angles is maximized.

The GSA considers the total reflectance ($R_I$); the polarized reflectance ($R_{\text{p}} = \sqrt{R_U^2 + R_Q^2}$); and also the degree of linear polarization (DoLP $= R_{\text{p}}/R_I$), which in RSP-like instruments is measured at a much higher accuracy than $R_{\text{p}}$ (Knobelspiesse et al., 2012; Cairns et al., 1999).

## 2.2 Global sensitivity analysis formalism

To illustrate the pitfalls of conventional sensitivity studies applied to hyperdimensional state spaces, Fig. 2 shows the sensitivity of $R_I$, $R_{\text{p}}$, and DoLP to $\text{AR}^{\text{T}}$ in the SWIR (see also Ottaviani, 2022), for two different values of $r_{\text{eff}}^{\text{T}}$. For $r_{\text{eff}}^{\text{T}} = 1280\,\mu\text{m}$ (solid lines) the signals are essentially unaffected by variations in $\text{AR}^{\text{T}}$; conversely, detectable differences arise for $r_{\text{eff}}^{\text{T}} = 56\,\mu\text{m}$ (dashed lines). Similarly, the DoLP grows with $r_{\text{eff}}^{\text{T}}$, but the curves for the two ARs are distinguishable only for small radii. Because the sensitivity to $\text{AR}^{\text{T}}$ depends on $r_{\text{eff}}^{\text{T}}$ and since the sensitivity of any parameter can similarly depend on the set of all other values kept fixed, it is necessary to use a method like the GSA to properly quantify the information content of the model.

The GSA framework relies on the computation of the so-called Sobol indices (Sobol, 1990). For a model function $g(X)$ of the $n$ state variables $X_1, \ldots, X_n$ (in our case, the parameters in Table 1), which is square-integrable over a parameter space $K^n$, there exists a functional decomposition in terms of a Haar wavelet basis given by

$$g(X) = g_{\text{o}} + \sum_{i=1}^{n} g_i (X_i) + \sum_{i<j}^{n} g_{i,j} \left( X_i, X_j \right)$$
$$+ \ldots + g_{1,2,\ldots,n} (X_1, X_2, \ldots, X_n). \tag{10}$$

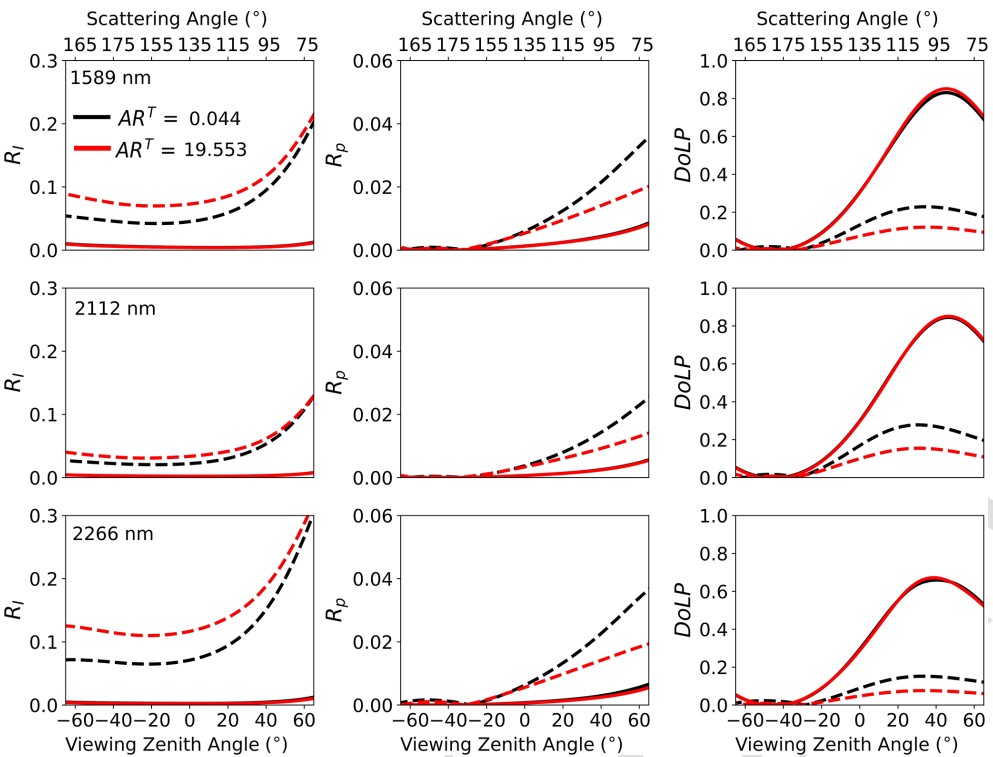

**Figure 2.** Sensitivity of $R_I$, $R_p$, and DoLP (columns) to $AR^T$, for $r_{eff}^T = 1280\,\mu m$ (solid lines) and $r_{eff}^T = 56\,\mu m$ (dashed lines). The remaining parameters are $D^T = 0.35$, $\rho^T = 0.1\,g\,cm^{-3}$, and $Z^T = 3\,cm$ (top layer), as well as $r_{eff}^B = 320\,\mu m$, $AR^B = 1.0$, $D^B = 0.35$, and $\rho^B = 0.3\,g\,cm^{-3}$ (bottom layer). The three rows are for the three SWIR wavelengths (1589, 2112, and 2266 nm).

The Haar wavelets form a basis for the space of all square-integrable functions ($L^2$). Because the simulated reflectances are bounded and smooth over the sample space and the sample space is compact, the model functions for our simulations are square-integrable (i.e., the decomposition in Eq. 10 exists). Squaring both sides of the equation, integrating over the whole parameter space, and using the orthogonality properties of the basis, one obtains

$$\int_{K^n} g^2(X)\mathrm{d}X - g_0^2 = \sum_{s=1}^{n}$$
$$\sum_{i_1 < \ldots < i_s}^{n} \int_{K^n} g_{i_1,\ldots,i_s}^2 \left(X_{i_1}, \ldots, X_{i_s}\right) \mathrm{d}X_{i_1}, \ldots, \mathrm{d}X_{i_s}. \quad (11)$$

If each variable $X_i$ is uniformly distributed over the parameter space $K^n$, the left-hand side exactly defines the variance of the model function $g(X)$:

$$\int_{K^n} g^2(X)\mathrm{d}X - g_0^2 = V_Y, \quad (12)$$

whereas the integrals

$$\int_{K^n} g_{i_1,\ldots,i_s}^2 \left(X_{i_1}, \ldots, X_{i_s}\right) \mathrm{d}X_{i_1} \ldots \mathrm{d}X_{i_s} = V_{i_1,\ldots,i_s} \quad (13)$$

are the variances of the functions $g_{i_1,\ldots,i_s}\left(X_{i_1}, \ldots, X_{i_s}\right)$. Combining the decomposition in Eq. (11) with Eqs. (12)

and (13) gives the decomposition of the total variance:

$$V_Y = \sum_{i=1}^{n} V_i + \sum_{i<j}^{n} V_{i,j} + \sum_{i<j<k}^{n} V_{i,j,k}$$
$$+ \ldots + V_{1,\ldots,n}. \quad (14)$$

Each $V_i$ term in the first sum corresponds to the "main-effect" contribution of the variable $X_i$ to the model output. The $V_{i,j}$ terms quantify the pairwise interactions between $X_i$ and $X_j$, $V_{i,j,k}$ the triplet-wise interactions among $X_i$, $X_j$, and $X_k$, and so on (Saltelli et al., 2008). Dividing both sides of Eq. (14) by the total variance, one obtains

$$1 = \sum_{i=1}^{n} S_i + \sum_{i<j}^{n} S_{i,j} + \sum_{i<j<k}^{n} S_{i,j,k}$$
$$+ \ldots + S_{1,\ldots,n}, \quad (15)$$

where the "total-effect" Sobol index for the parameter $X_j$,

$$S_{T_j} = 1 - \sum_{j \notin \{i_1,\ldots,i_s\}} S_{i_1,\ldots,i_s}, \quad (16)$$

quantifies the complete contribution of $X_j$ to the total variance over the entire parameter space, both directly and through interactions among parameters. The Sobol indices are calculated at each angle with the Python software package SALib (Herman and Usher, 2017; Iwanaga et al., 2022), which uses the quasi-Monte Carlo estimators presented in

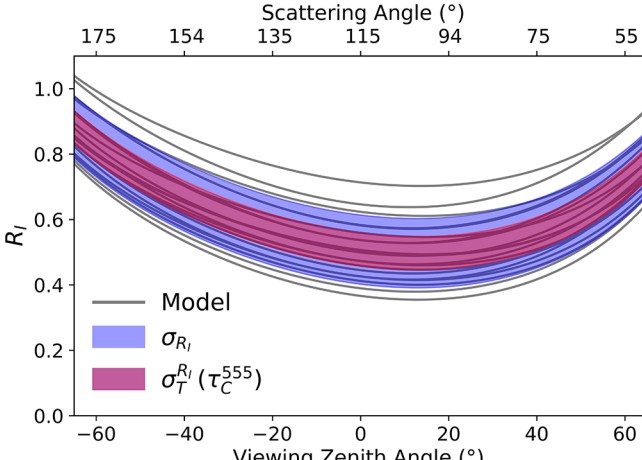

**Figure 3.** Total reflectance $R_I$ (gray lines) at 555 nm simulated for 20 random combinations of the state parameters, as a function of the viewing zenith angle along the principal plane (SZA: 65°). The total standard deviation for this ensemble of curves $\sigma_{R_I}$ is shown by the blue-shaded region, and the red-shaded region corresponds to the absolute total-effect Sobol index $\sigma_T^{R_I}$ associated with variations in $\tau_C^{555}$. The fact that $\sigma_T^{R_I}$ covers a large portion of $\sigma_{R_I}$ indicates that a large amount of the variation in $R_I$ is due to $\tau_C^{555}$.

Saltelli et al. (2010). To compare the indices at different wavelengths, they are converted into absolute quantities by multiplying by the total variance at that angle:

$$\sigma_{T_j}^Y = \sqrt{S_{T_j} \cdot V_Y}. \tag{17}$$

A visual interpretation of the Sobol indices is given in Fig. 3, where the gray curves show the total reflectance at 555 nm output by the radiative transfer model for 20 random combinations of input parameters, computed for a solar zenith angle (SZA) of 65°. The region shaded in blue represents
the total variance of the model curves and the area in red the absolute total-effect Sobol index $\sigma_T^{R_I}$ relative to $\tau_C^{555}$. A comparison between $\sigma_{R_I} = \sqrt{V_{R_I}}$ and $\sigma_T^{R_I}$ reveals that a significant portion of the total variance of the model is due to variations in $\tau_C^{555}$.

The sensitivities are evaluated against the $1\sigma$ uncertainty corresponding to the square root of the diagonal elements of the measurement error covariance matrix (Knobelspiesse et al., 2012). Full measurement covariance matrices, which include off-diagonal elements expressing cross-correlation ef-
fects, can be hard to assess (Gao et al., 2023). In line with other studies that consider the uncertainty in multi-angle polarimeters (Hasekamp, 2010; Lebsock et al., 2007; Stamnes et al., 2018; Ottaviani et al., 2012), we also neglect such elements that in any case do not affect our main conclusions.
The light gray areas in Fig. 4 (see also discussion in the next section) correspond to a 3 % (0.5 %) radiometric (polarimetric) accuracy, nominally achievable by modern spaceborne sensors like the polarimeters launched on board PACE.

The dark gray areas correspond instead to the higher accu-
30 racy of RSP-like sensors (1.5 % radiometric and 0.2 % polarimetric). Note that the uncertainty in $R_p$ is rather different in the two cases, since the error model used for the areas in light gray includes a term proportional to $R_I^2$. This term is a result of a filter-wheel-type design, where the measurements
needed to compile the Stokes vector of any given scene are not acquired simultaneously (Dubovik et al., 2019; Knobelspiesse et al., 2012), and is large over bright surfaces, as is the case for snow in the VIS–NIR.

If $\sigma_{Ti}$ is less than the threshold at all angles, the associ-
40 ated parameter is ruled out as a meaningful contributor to model variance and excluded from the plot. The SALib package also gives relative confidence intervals for each Sobol index, which are converted to absolute confidence intervals around $\sigma_{Ti}$. A sample size $m = 2^{16}$ (specified as a power of
45 2 as required by the software package) was chosen for a total of $(n+1)m = 1\,048\,575$ ⬛TS1 runs (where $n = 14$ is the number of parameters), which ensures that for all $\sigma_T$ curves which lie above the uncertainty threshold, the confidence interval also lies above the uncertainty threshold.

As discussed below, values of $\sigma_{Ti}$ above the uncertainty threshold do not necessarily guarantee retrievability, which is impacted by model uncertainty and other sources of error unaccounted for in the covariance matrices used here. In this respect, the retrievals can be considered to be a best-case
scenario (Rodgers, 2000; Knobelspiesse et al., 2012).

## 3 Results and discussion

### 3.1 Pure snow

As a first example of the application of the GSA, in Fig. 4 we show the angular variations in the absolute total Sobol in-
60 dices computed for a pure-snow scene. Although such an idealized case is not commonly encountered in real-world scenarios, this exercise is useful for isolating and understanding the sensitivity to the parameters describing the ice crystals; the effects of LAPs are then discussed in the next section.
In both cases, we consider optically semi-infinite snowpacks since the focus of this paper is on the remote sensing of snow; heterogeneous pixels constitute an added layer of complexity and will be the subject of future studies.

The results are for principal-plane observations and SZA = 65°. Note that the top $x$ axis in the first row reports
the conversion to scattering angles, which is useful for connecting scanning viewing zenith angles (VZAs) to forward-scattering and backscattering directions in the discussions that follow. The different wavelengths are in different rows, and only state parameters exhibiting sensitivity over the $1\sigma$
instrument uncertainty threshold are included. The GSA correctly captures the known dependence of the reflectance on the top-layer grain size in the SWIR (increasing with viewing zenith angle in the forward-scattering half-plane; see also

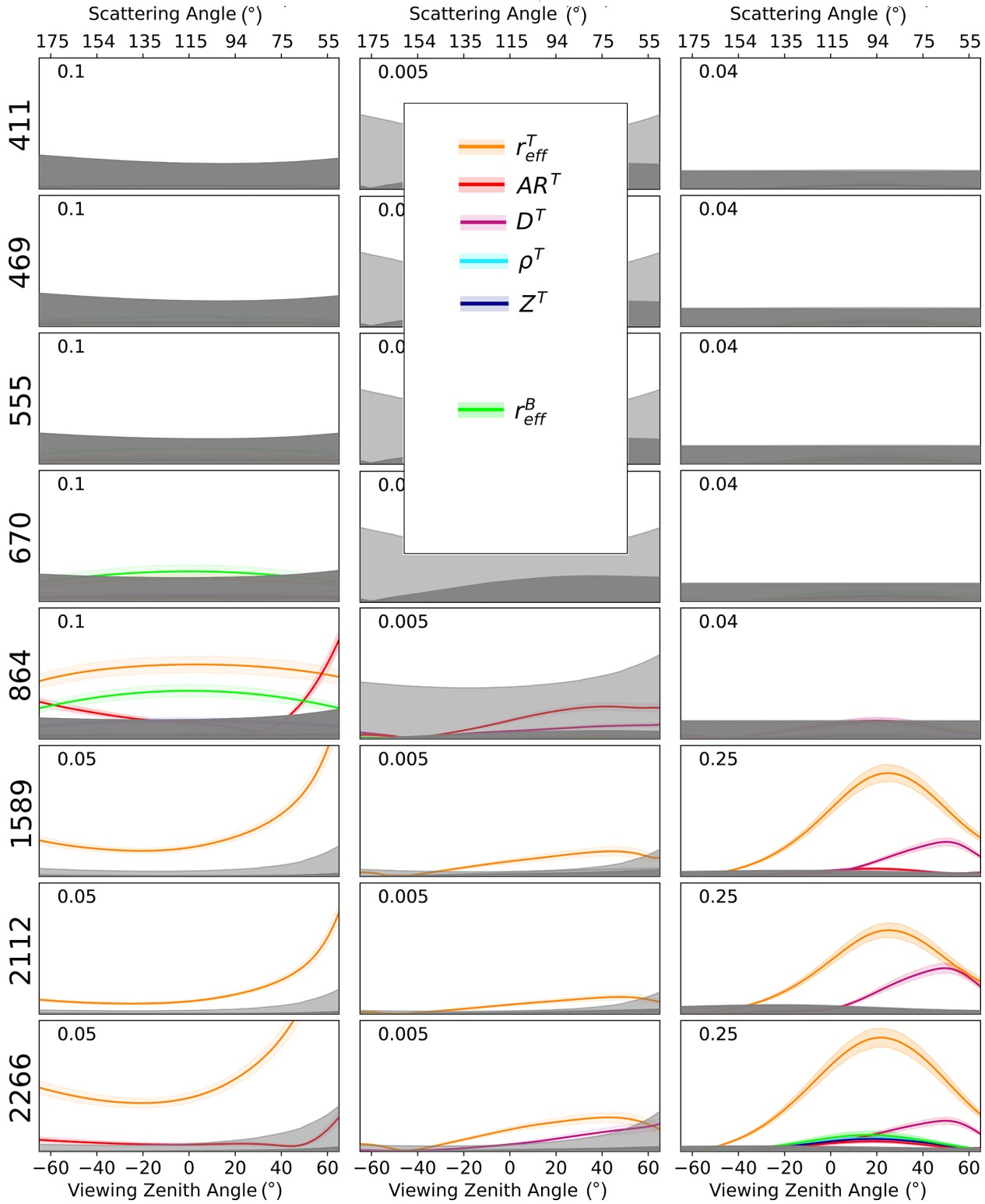

**Figure 4.** Absolute total Sobol index $\sigma_T$ for $R_I$, $R_p$, and DoLP (different columns) as a function of the viewing zenith angle along the principal plane and for a SZA of 65°, computed for a pure snowpack under a clear atmosphere ($C_{BC}^T = C_{BC}^B = 0$ ppmw, $\tau_C^{555} = 0$). The indices with their 90 % confidence intervals are only given for parameters with variance above the instrument uncertainty thresholds.

Fig. 2) due to the large absorption by ice at these wavelengths (Wiscombe and Warren, 1980; Dang et al., 2016), which is exploited in retrieval schemes (Stamnes et al., 2007; Painter et al., 2009, 2003; Nolin and Dozier, 1993). The lack of sensitivity of $R_I$ to grain size in the VIS is explained by the fact that pure snow is highly reflective (i.e. non-absorbing) at visible wavelengths, which therefore are normally not used for size retrievals.

The exact penetration depth of each wavelength in the snowpack depends on grain size, snow density, and impurity content. For pure snow of typical densities and grain sizes, the VIS wavelengths penetrate deeper (10–25 cm) than NIR (3–20 cm) and SWIR (0–2 cm) wavelengths (Kokhanovsky, 2022; Libois et al., 2013; Li et al., 2001). Multiple scattering quickly randomizes polarization as the incident light penetrates into a dense medium (van Diedenhoven et al., 2013), explaining why no detectable sensitivity to any of the parameters is found for $R_p$ and DoLP in the VIS within the considered uncertainties. The situation is different in the SWIR, again because of the strong ice absorption. The very limited penetration depth allows the polarization signatures determined by the single-scattering properties of the ice crystals at the very top of the snowpack to be preserved, especially for observations of the DoLP, which are typically achieved with higher measurement accuracy. Besides the evident sensitivity to $r_{eff}^T$ and $D^T$ in this wavelength regime, an interesting result concerns the 2266 nm RSP channel, which seems to access detectable sensitivity to $AR^T$ not present for the nearby MODIS channel at 2112 nm. Furthermore, the reduced uncertainty in RSP-like sensors reveals sensitivity of the polarized reflectance measured at 864 nm to $D^T$ and $AR^T$ (Ottaviani et al., 2015).

All simulated measurements are insensitive to the mixing proportion ($f$) of columns and plates in both layers, which can be explained by the similarity of the scattering properties for crystals with reciprocal ARs (see Fig. A3). Because of its large physical thickness, the bottom layer is optically semi-infinite regardless of $\rho^B$. Directional changes in the light scattered downward in response to variations in $D$ and AR do not prevent its fast extinction, and the fraction of upwelling photons supplied by the bottom layer stays pretty constant so that $R_I$ is insensitive to $\rho^B$.

Finally, we note that the LUT includes random selections for the thickness of the top layer and for the snow density in both layers. Other than the minimal information contained in the DoLP at 2266 nm (blue and cyan curves, nearly overlapped), the Sobol indices reveal that these parameters cannot be independently retrieved. The optical depth of the top layer largely determines the observed signal but is proportional to the product of $\rho^T$ and $Z^T$, which is invariant when one parameter is divided by the same factor used to multiply the other. The same argument is even more valid for the semi-infinite bottom layer, which prohibits passive optical measurements from accessing information on its thickness or density.

The results of the GSA are particularly useful for informing the choice of parameters to be included in the state vector of inverse retrievals. As an example, we generated synthetic TOA observations with the RT code, including random noise added according to the specifications of different sensors. As explained in Sect. 2, the snowpack consists of a mixture of crystals ($f^T = f^B = 0.5$). Fresher snow (smaller grains) is simulated in the top layer ($r_{eff}^T = 150\,\mu m$, $Z^T = 3\,cm$, $\rho^T = 0.2\,g\,cm^{-3}$, and $AR^T = 0.05$ for plates; corresponding $1/AR^T = 20$ for columns; and $D^T = 0.3$ as found by Ottaviani et al., 2015). More compact, larger, and rounder grains are located in the bottom layer ($r_{eff}^B = 250\,\mu m$, $\rho^B = 0.30\,g\,cm^{-3}$, $AR^B = 0.15$ for plates and 6.67 for columns, $D^B = 0.40$), which is optically semi-infinite ($\tau \approx 2000$). For the reasons given at the end of Sect. 3, the thickness of the top layer and density of both layers were not included in the set of retrievables and, together with the other parameters excluded from the state vector, are constrained to the values used to generate the synthetic observations.

Using the LMFit Python library (Newville et al., 2014), a Levenberg–Marquardt nonlinear least-squares optimal estimation scheme (Levenberg, 1944; Marquardt, 1960) was then implemented to retrieve the input parameters. We first consider the configuration of an RSP-like instrument. In satellite imagery, every pixel is characterized by its own set of viewing zenith and azimuth angles. The RSP is instead a scanner, and we chose the principal plane as a scanning direction because it guarantees that the viewing geometries span the largest range of scattering angles.

To highlight the merits of polarization, we compare retrievals that consider the measurement vectors to be the simulated (i) total reflectance, (ii) total reflectance and polarized reflectance, and (iii) total reflectance and DoLP. These retrievals are repeated considering VIS, VIS–NIR, or VIS–NIR–SWIR wavelengths. The SWIR combination consists of 1589 and 2266 nm. All available viewing angles (150 measurements for RSP, roughly between ±70° TS2) are used for the total reflectance. Figure 4 shows that the DoLP and $R_p$ are largely unaffected by variations in any of the parameters (except $r_{eff}^T$) for angles in the backscattering half-plane. If $R_p$ or DoLP for these angles is included in the retrieval, the minimization algorithm attempts to fit very small changes in $R_p$ or the DoLP, which cannot be distinguished from the noise, and the retrieval quality for all parameters (other than $r_{eff}^T$) degrades. CE1 Consequently, we subsampled the measurements of $R_p$ and DoLP to positive viewing zenith angles, which in our reference system correspond to the forward-scattering half-plane.

Figure 5 summarizes the values of the state parameters and their uncertainty obtained from the inversion. The solid lines represent the "true" values used in the forward simulations. In contrast, the dashed lines are the initial guess for each parameter, randomly sampled within the bounds listed in Ta-

ble 1. The retrievals were repeated a few times to test the stability of the results against the different initial guesses.

In this simplistic scenario of a pure snowpack and a clear atmosphere, Fig. 4 indicates that $R_I$, $R_p$, and DoLP in the VIS are insensitive to all parameters (except for $R_I$ to $r_{\mathrm{eff}}^{\mathrm{B}}$ at 670 nm), and retrievals attempted with these data alone are unsuccessful. The addition of NIR measurements of $R_I$ and $R_p$ (if the accuracy of the latter matches RSP levels) gives access to information on grain shape and microscale roughness, as confirmed with real data (Ottaviani et al., 2012, 2015).

The retrieval becomes optimal when polarization capabilities in the SWIR are also included. Using $R_I + R_p$ or $R_I + \mathrm{DoLP}$ leads to similar performances, with the inversions converging to the true values within the error bars for all parameters except for $r_{\mathrm{eff}}^{\mathrm{B}}$, which is in any case retrieved moderately well.

We next turn our attention to retrievals simulated for instruments with spectral and angular configurations different from those of the RSP. A MODIS-like sensor is mimicked by considering mono-angle measurements of total reflectance in the VIS–NIR–SWIR. The addition of 16-angle polarimetric measurements in the VIS–NIR mimics POLDER. The viewing geometries for the simulations are taken from actual data collected over Greenland for a pixel near Summit Station (72° N, 39° W) on 8 April 2007 and for SZA $\approx 65°$, which corresponds to the SZA used to generate Fig. 4. All available angular measurements are considered for the total reflectance in the VIS–NIR, subsampled to the same range as in the RSP-like case.

The results obtained using these two different instrument configurations are compared to those of the RSP in Fig. 6. Retrievals using MODIS-like data can recover the top-layer grain size (Stamnes et al., 2007; Hori et al., 2007; Painter et al., 2009, 2003; Nolin and Dozier, 1993) but fail to resolve all other parameters in a satisfactory manner. Information on grain shape is still accessible to the MODIS NIR channel but for the viewing zenith angle away from nadir, which exemplifies the utility of multi-angular measurements. Observations of $R_p$ contain information on $D^{\mathrm{T}}$ in the NIR, although the larger uncertainty assigned to the simulated spaceborne measurements limits the retrieval quality compared to the RSP-like case. Finally, we note that all these retrievals are robust against different choices of the initial guess for each parameter.

## 3.2 Scenes containing snow impurities and aerosols

In this section, we expand the analysis to include more realistic scenes characterized by the presence of LAPs both in the snowpack and as atmospheric aerosols. We also consider different SZAs.

Light-absorbing impurities in the snow are often found in very significant amounts in North America, China, and the Arctic (Warren, 2019). In Greenland, especially on the plateau, the concentrations are typically much smaller and

therefore difficult to detect via remote sensing (Warren, 2013). Since they anyway have a significant impact on snow visible albedo (Warren and Wiscombe, 1980; Dang et al., 2016), their accurate determination is especially important for climate modeling (Antwerpen et al., 2022; Wang et al., 2020; Alexander et al., 2014; Ryan et al., 2019). To target these challenging retrievals, the results of the GSA are computed for maximum LAP loads of 0.4 for $\tau_{\mathrm{C}}^{555}$ and 1 ppmw for $C_{\mathrm{BC}}^{\mathrm{T}}$ and $C_{\mathrm{BC}}^{\mathrm{B}}$ (Fig. 7), where the subscript "BC" indicates the specific type of black-carbon LAP considered in this paper, with fixed microphysical and optical properties. More sporadic events like thick burning plumes or exceptionally dirty snow are addressed in Appendix A, where the same calculations are repeated with extended ranges of $\tau_{\mathrm{C}}^{555}$ (up to 1.2) and $C_{\mathrm{BC}}^{\mathrm{T}}$ and $C_{\mathrm{BC}}^{\mathrm{B}}$ (up to 10 ppmw). At these higher LAP amounts, the sensitivity of polarimetric measurements in the SWIR to aerosol optical depth is even more pronounced (see Fig. A1 and related discussion).

The sensitivity to top- and bottom-layer parameters can be interpreted in terms of the penetration depths discussed in Sect. 3.1. The top-layer thickness ranges between 1 and 5 cm in the LUT, explaining why the DoLP is (weakly) sensitive to $r_{\mathrm{eff}}^{\mathrm{B}}$ in the SWIR at 2266 nm but not at 1589 and 2112 nm, for which the ice absorption is slightly larger.

The total reflectance in the VIS–NIR shows the expected sensitivity to LAPs. The added sensitivity to $\tau_{\mathrm{C}}^{555}$ reflects the additional information provided by polarization. The fact that the polarized reflectance and the DoLP measured in the SWIR are insensitive to the concentration of LAPs in the snowpack implies that polarimetry can also be exploited to (i) extend heritage aerosol retrievals performed over other land surfaces to snow surfaces and (ii) inform the vertical partitioning of LAPs between the atmosphere and the surface (Ottaviani, 2022).

To examine the dependence on solar illumination, Fig. 8 was produced for a SZA of 45°. At this smaller SZA, the angles of maximum $\sigma_{\mathrm{T}}^{\mathrm{DoLP}}$ shift toward larger viewing angles. Additionally, $\sigma_{\mathrm{T}}^{R_I}$ in the VIS–NIR increases for all parameters. These differences are only minor; the lack of appreciable changes with SZA, at least in this range typical of high latitudes, is therefore attractive in the context of remote sensing applications.

In the VIS–NIR, $\sigma_{\mathrm{T}}^{R_I}$ exhibits an essentially flat behavior well above the detection thresholds at all viewing zenith angles for many of the parameters, with shallow maxima at around nadir except for $\mathrm{AR}^{\mathrm{T}}$ and $\tau_{\mathrm{C}}^{555}$. In the SWIR, $\sigma_{\mathrm{T}}^{R_I}$ and $\sigma_{\mathrm{T}}^{R_p}$ for $r_{\mathrm{eff}}^{\mathrm{T}}$ and $\tau_{\mathrm{C}}^{555}$ peak at the largest viewing zenith angles. The DoLP now includes sensitivity to $D^{\mathrm{T}}$, still occurring in the forward-scattering half-plane but with peaks at smaller angles. Multi-angle polarization measurements can therefore greatly supplement those of total reflectance, especially when retrieving parameters that express marked angular differences in $\sigma_{\mathrm{T}}$.

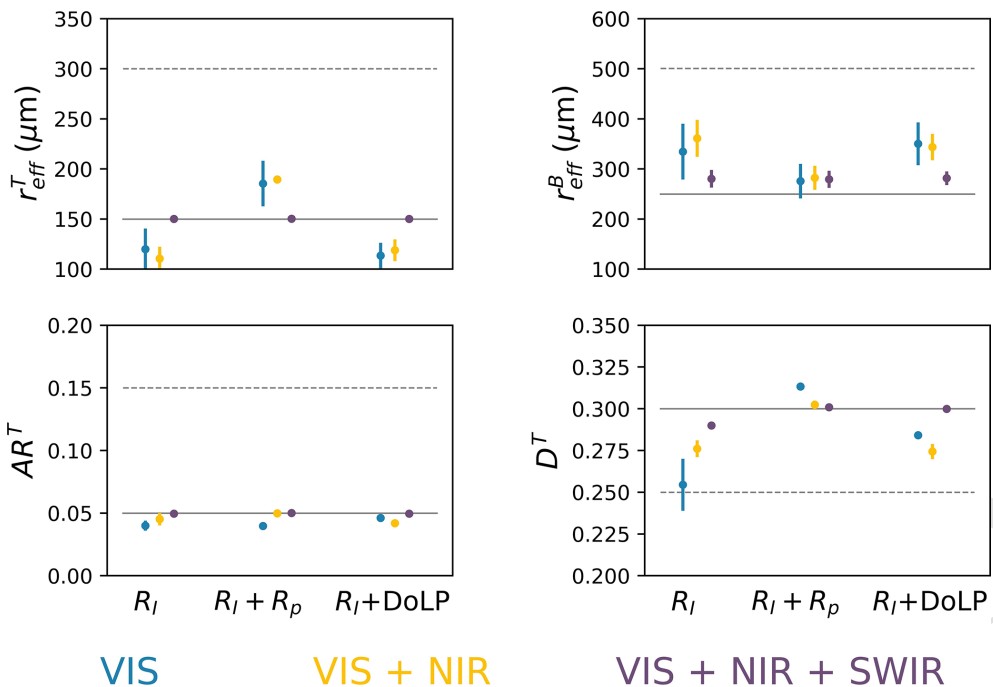

**Figure 5.** State parameters (different panels) retrieved from the inversion of RSP-like TOA observations, generated for a SZA of 65° along the principal plane. The scene consists of a pure snowpack and a clean atmosphere (see text). The inversions are repeated with and without the inclusion of polarization and at different wavelength combinations. The solid and dashed gray lines are the true values and the initial guesses for each parameter.

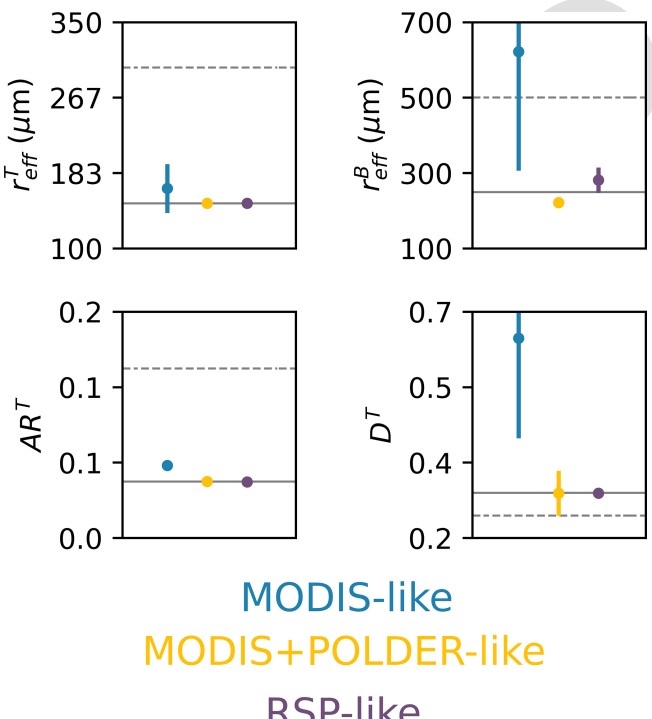

**Figure 6.** Similar to Fig. 5 but for inversion results of simulated MODIS-, MODIS + POLDER-, and RSP-like observations.

Figure 9 provides an alternative display of the information contained in Figs. 7 and 8, for a SZA of 65° (top panel) and SZA of 45° (bottom panel). These heatmaps can aid in the choice of appropriate viewing geometries and channel combinations when designing retrieval algorithms and observational strategies. The intensity of each cell's color is proportional to the maximum value of $\sigma_{\mathrm{T}}^{R_I}$ (left columns), $\sigma_{\mathrm{T}}^{R_{\mathrm{p}}}$ (middle columns), and $\sigma_{\mathrm{T}}^{\mathrm{DoLP}}$ (right columns) across all VZAs, and the number reports the angular location of these maxima. Numbers close to 0 represent nadir-looking directions, and large positive angles correspond to the forward-scattering directions (see top $x$ axis in Figs. 7 and 8). It is evident that measurements in the forward-scattering half-plane are sensitive to the properties of aerosols and the top snow layer, while nadir-looking geometries favor the determination of parameters deeper in the snowpack. It is also clear that the addition of accurate polarimetric measurements in the VIS–SWIR benefits the retrieval of aerosol and surface properties, especially if they are present at multiple angles.

The simulated retrievals in Sect. 3.1 were duplicated with the inclusion of LAPs. The impurity concentration in the snowpack was set to $C_{\mathrm{BC}}^{\mathrm{T}} = C_{\mathrm{BC}}^{\mathrm{B}} = 2.0 \times 10^{-3}$ ppmw, typical of the Greenland plateau (Warren, 2019). The atmosphere contains aerosols with $\tau_{\mathrm{C}}^{555} = 0.10$. Figure 10 summarizes the state parameters and their uncertainty for each type of re-

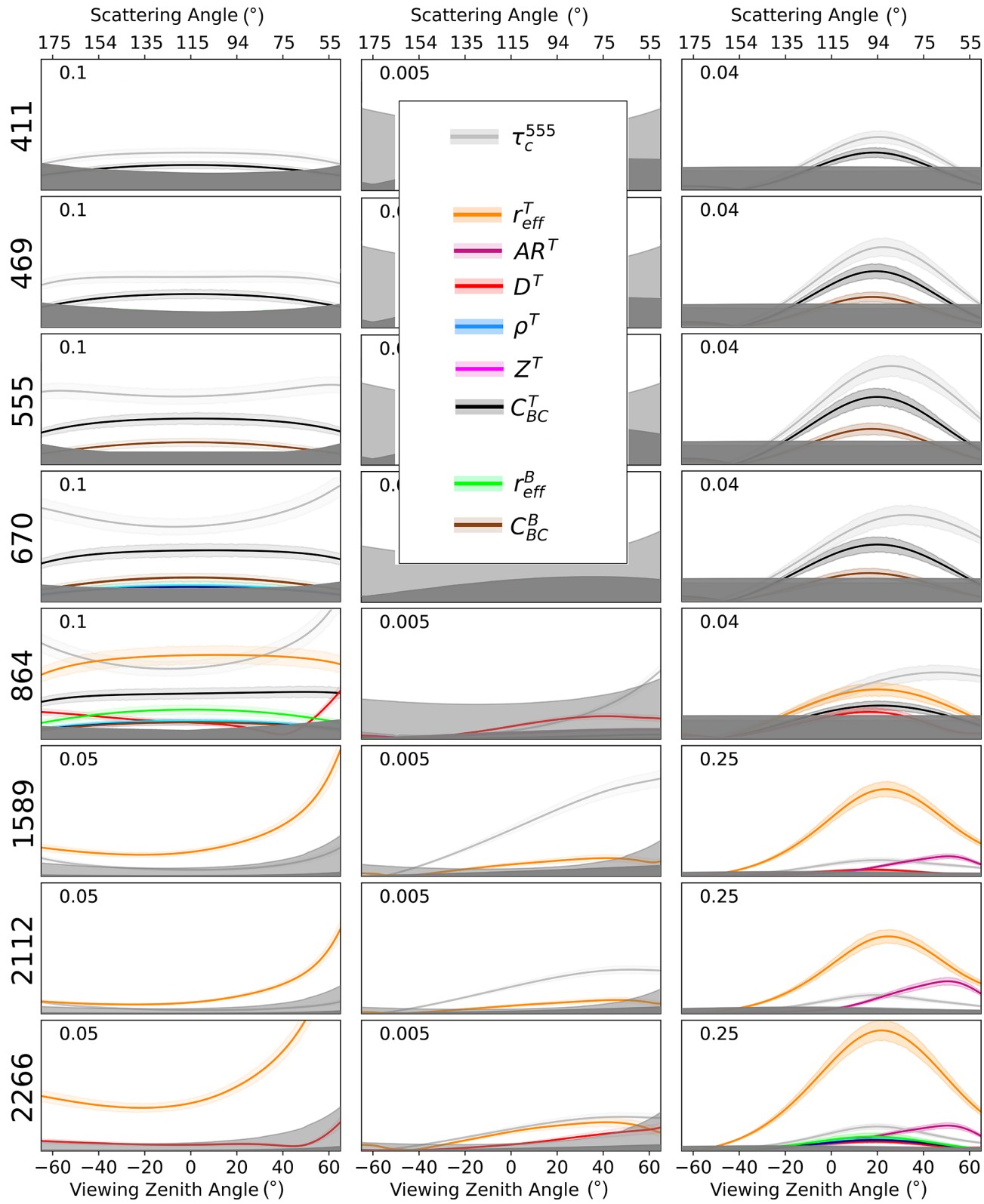

**Figure 7.** Same as Fig. 4 but for variable amounts of LAPs within the snowpack and in the atmosphere above.

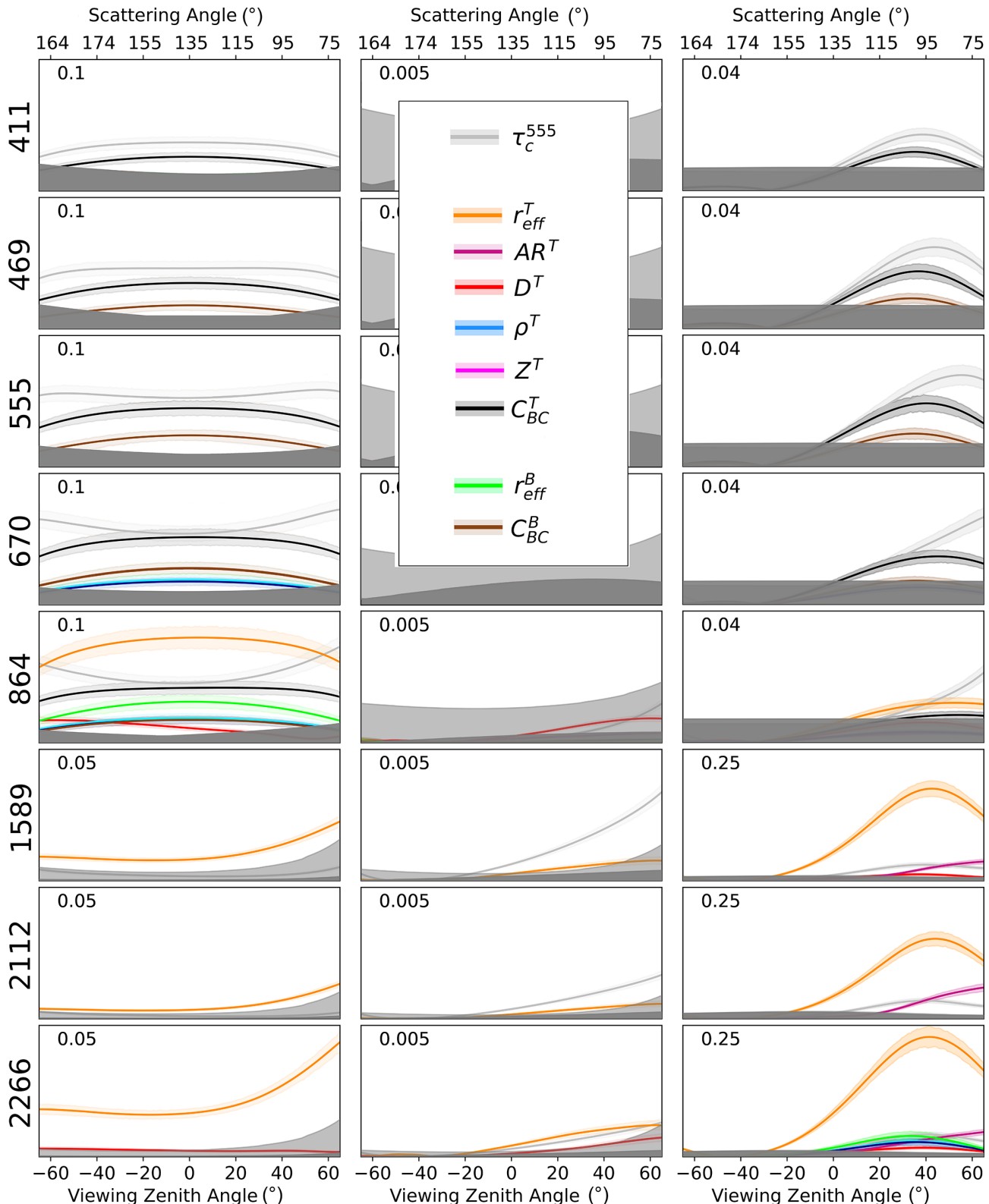

**Figure 8.** Same as in Fig. 7 but for a solar zenith angle of 45°.

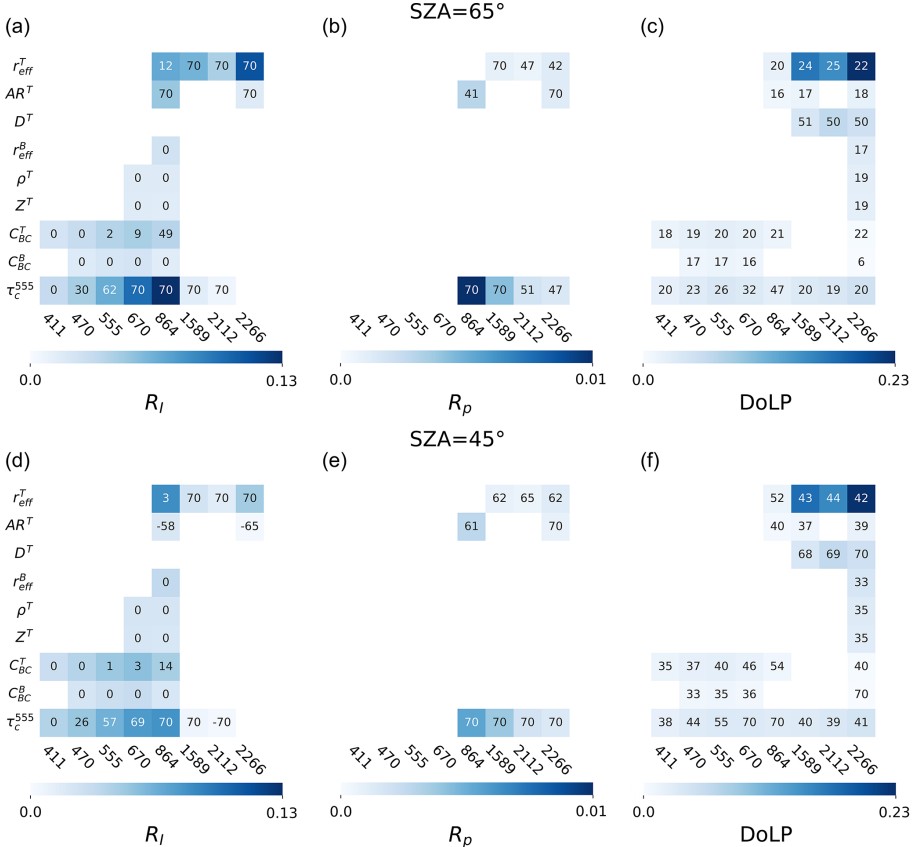

**Figure 9.** Heatmaps of the maximum value of $\sigma_{\mathrm{T}}^{R_I}$, $\sigma_{\mathrm{T}}^{R_{\mathrm{p}}}$, and $\sigma_{\mathrm{T}}^{\mathrm{DoLP}}$ across all viewing angles, for each parameter (row) and wavelength (column) combination. The top panel is for a SZA of 65° and the bottom panel for a SZA of 45°. The number in each box is the viewing zenith angle at which the maximum occurs. Only parameters with sensitivity over the uncertainty thresholds are included.

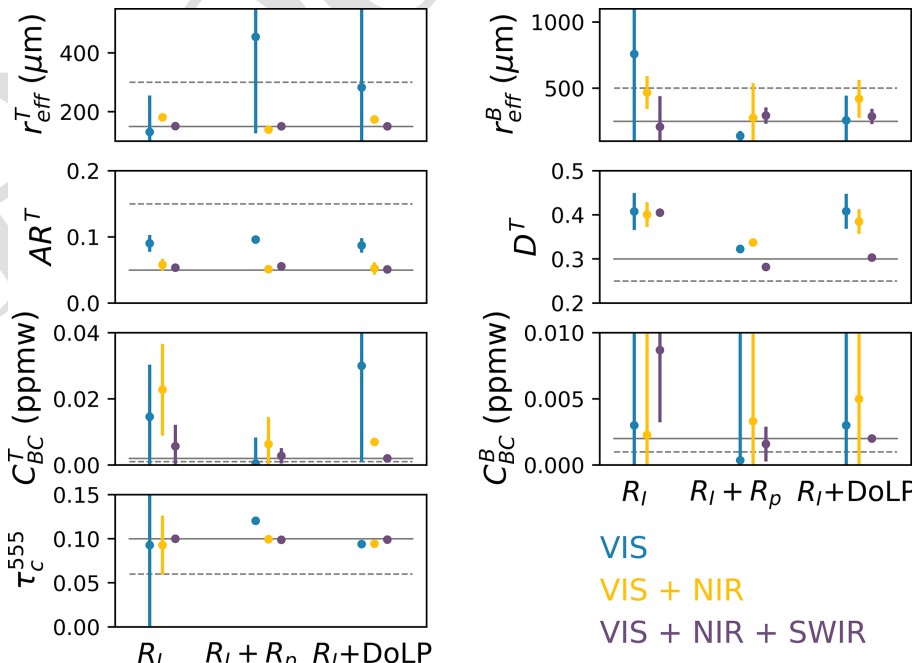

**Figure 10.** Same as Fig. 5 but with LAPs included in the retrievals.

trieval, at different measurement vector and wavelength combinations.

Retrievals which use $R_I$ + DoLP from only VIS and NIR channels are in general not successful and fail to determine the vertical distribution of impurities because they are challenged by the simultaneous sensitivity to multiple parameters. Improvements are observed when measurements in the SWIR are included due to selective sensitivity to $r_{\mathrm{eff}}^{\mathrm{T}}$ and $\tau_{\mathrm{C}}^{555}$.

Unsurprisingly, the best performance is achieved by including both total reflectance and DoLP in the VIS–NIR–SWIR. The uncertainties decrease by an order of magnitude when compared to measurements of total reflectance only, confirming that polarimetric measurements in the SWIR are valuable for determining the vertical partitioning of LAPs. Initializing the inversion with first guesses close to the true values helps to decrease the uncertainty in $C_{\mathrm{BC}}^{\mathrm{T}}$ and $C_{\mathrm{BC}}^{\mathrm{B}}$ retrieved from the VIS–NIR–SWIR combination of $R_I$ and $R_p$, but the best performance is still obtained using $R_I$ + DoLP.

We next repeat the retrievals using MODIS and MODIS + POLDER-like measurements. For the latter, we use the DoLP in the VIS–NIR in place of $R_p$ because the DoLP manifests detectable sensitivity to LAPs. Figure 11 shows that in the MODIS-like case the inversion struggles to retrieve all parameters except $r_{\mathrm{eff}}^{\mathrm{T}}$, and the considerable uncertainties can severely impact the accuracy of derived albedo estimates. The addition of multi-angle polarimetric data enables a better determination of all parameters, even at lower angular resolution than the RSP's (Wu et al., 2015), since the angular radiative behavior of the different system components is rather smooth. One exception concerns the vertical profile of impurities in the snow: only when assuming a uniform concentration in the two layers is the retrieval successful (not shown).

## 4    Conclusions

The information content of polarimetric simulations over snow scenes was evaluated using a global sensitivity analysis (GSA) method, which accounts for the correlated sensitivity to model parameters across the entire parameter space. A comprehensive look-up table (LUT) was created with an advanced vector radiative transfer model, spanning wavelengths from the VIS to the SWIR. The snow–atmosphere system is vertically resolved and accounts for the presence of light-absorbing particulates (LAPs) both embedded in the snow and aloft in the atmosphere above the snowpack. The Sobol indices computed from the LUT are the primary metrics for the GSA and show the expected sensitivity of total reflectance in the VIS–NIR to LAPs and in the SWIR to snow grain size. In contrast to measurements of total reflectance only, polarimetric measurements inform the vertical distribution of LAIs in the system thanks to differential sensitivity present especially in the SWIR. Retrievals of grain shape

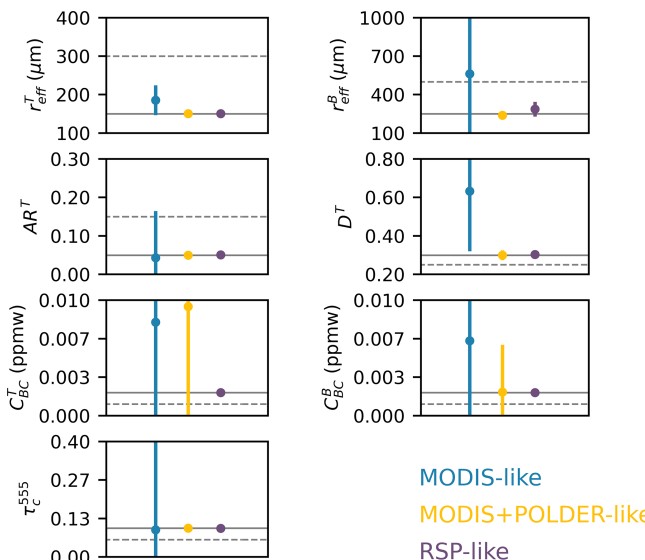

**Figure 11.** Same as Fig. 6 but for a scene containing LAPs in the snow and in the atmosphere.

from polarimetric measurements in the NIR can be improved by the addition of SWIR channels, leading to better estimates of the asymmetry parameter and, in turn, of the albedo in climate models. The angular dependence of the sensitivity (especially of the DoLP) emphasizes the advantages of exploiting sensors with multi-angular capabilities. The findings are largely independent of the solar zenith angle at least at high latitudes, an additional advantage for remote sensing applications.

The information content analysis was used to inform the choice of state parameters to be retrieved in sample Levenberg–Marquardt inversions, which were tested on synthetically generated polar scenes for different instrument configurations. The retrievals indeed confirm that mono-angle measurements of total reflectance in the VIS–SWIR (i.e., MODIS) can adequately resolve the grain size in the top layer, while access to more complex descriptors for the snow grains (in our case the aspect ratio and microscale roughness of hexagonal prisms) is achieved by the addition of multi-angle, polarimetric measurements in the NIR–SWIR. Such observations also make it possible to differentiate LAPs in the snow from absorbing aerosol layers, a task that can improve the characterization of processes like aerosol deposition in climate models and, again, albedo simulations.

The findings generally promote the use of the DoLP over the polarized reflectance and indicate that observations from the VIS all the way to the SWIR minimize the uncertainties when attempting to distinguish impurities in snow from absorbing aerosols. The GSA can be extended to LUTs that consider a whole suite of aerosol optical properties, region-specific impurity amounts, and more elaborate mixing schemes (Tanikawa et al., 2020) or optically thin

snowpacks with different underlying land cover types. However, the methods and results outlined in this paper provide cryospheric scientists with guidelines for selecting appropriate viewing geometries during data collection and for developing advanced retrieval algorithms applied to airborne and spaceborne data over snow. This perspective is particularly exciting considering the higher accuracies enabled by recent technological progress, like those of the polarimeters on PACE and 3MI.

# Appendix A

This Appendix discusses a few aspects that could not be included directly in the main text of the paper without unnecessarily interrupting the flow.

Figure A1 is the same as Fig. 7 but considers a larger range of aerosol optical depth (up to 1.2) and impurity density (up to 10 ppmw), which can occur as a result of burning biomass plumes or extremely "dirty" snow. With higher concentrations of LAPs, the absolute Sobol indices for the visible and NIR wavelengths increase (see Eq. 17) because BC absorption causes large variations in the total reflectance and, to a more limited extent, in the polarized reflectance, especially at larger viewing zenith angles (see also Fig. 3 in Ottaviani, 2022). Larger DoLP signals also lead to a minor increase in the uncertainty threshold, but the list of parameters with identified sensitivity in Fig. 7 remains the same.

A curious aspect concerns the growing sensitivity to $r_{eff}$, as the concentration of snow impurities increases, as shown with the sensitivity studies performed over specific "slices" of the LUT in Fig. A2. Size measurements are achieved by exploiting absorption. Pure snow is highly reflective in the VIS–NIR, and these wavelengths cannot be exploited to retrieve the effective radius of the snow grains, as shown by the solid lines. However, for high concentrations of impurities in the snowpack, the total reflectance and DoLP in the VIS–NIR do show dependence on $r_{eff}^{T}$ (dashed lines). The reason for this behavior is that the impurities in the model are externally mixed with snow (Tanikawa et al., 2020) and occupy the empty spaces between grains; the absorption occurring in these negative spaces is therefore partly correlated with the dimension of the crystals.

The differences in the Sobol indices compared to Fig. 7 are only minor. Polarimetric measurements in the SWIR show selective sensitivity to $r_{eff}^{T}$, to $D^{T}$, and (even more prominently) to $\tau_{C}^{555}$. The general conclusions drawn in the main text therefore remain unaffected if a larger range of LAPs is considered. As reported in Sect. 3, the GSA finds no sensitivity to the column-to-plate fraction for mixtures of columns and plates with an aspect ratio (AR) of 1 / AR. This fact is explained by the very similar asymmetry parameters of grains with reciprocal aspect ratios (van Diedenhoven et al., 2014a), as shown in Fig. A3 for columns with $AR^{T} = 19.553$ (black) and plates with $AR^{T} = 1/19.553 = 0.051$ (red).

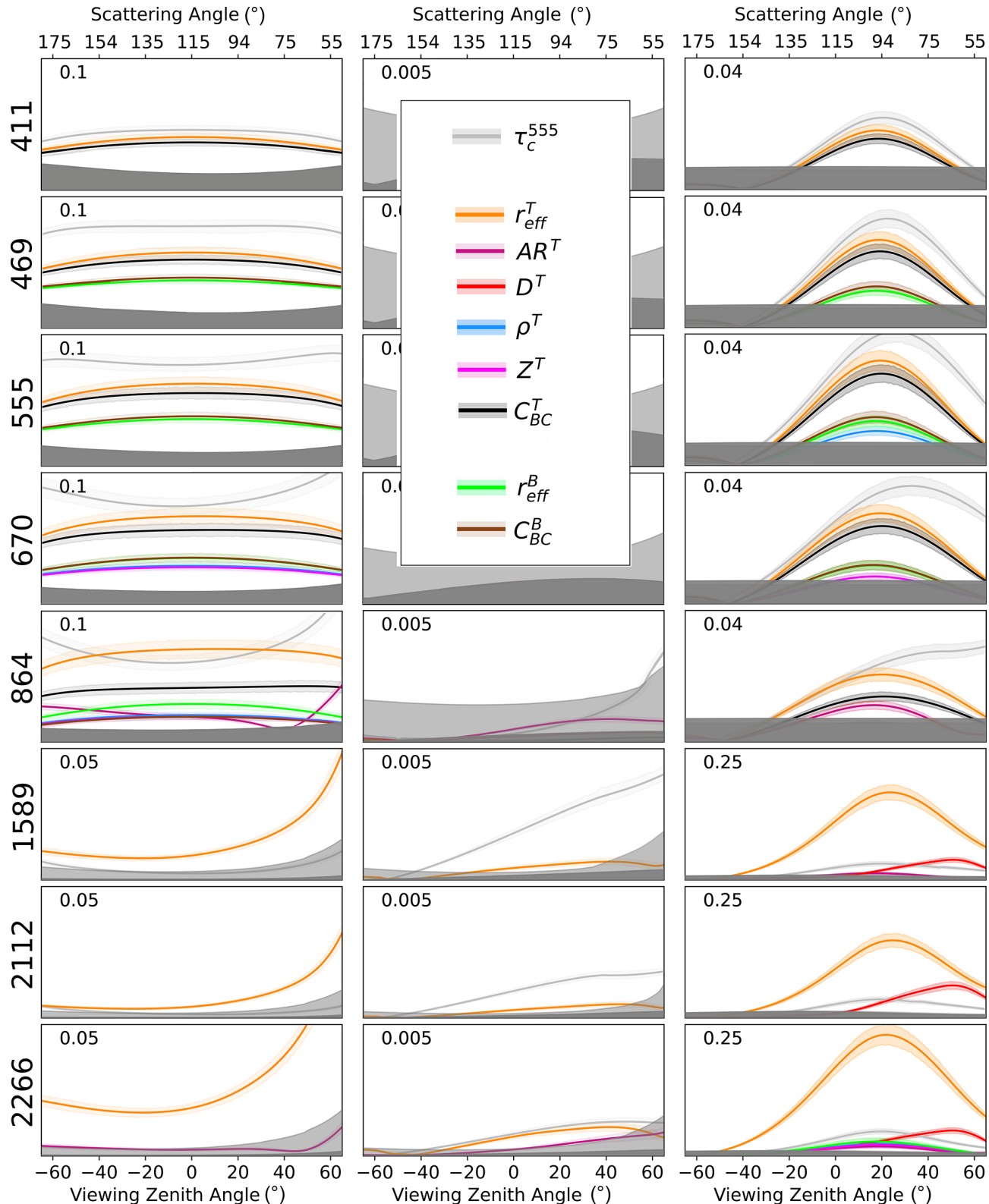

**Figure A1.** Same as Fig. 7 but for a larger range of aerosol optical depth (up to 1.2) and impurity density (up to 10 ppmw).

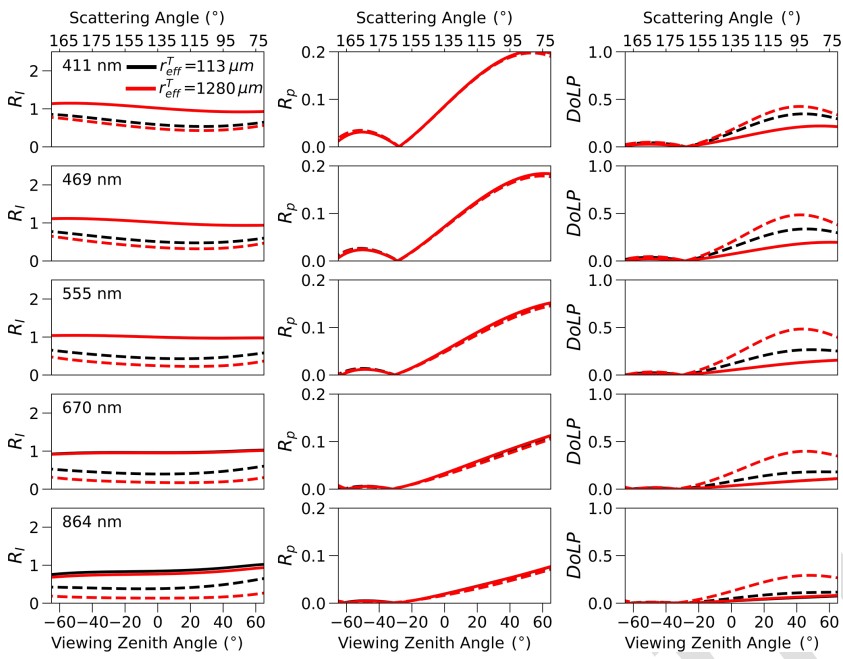

**Figure A2.** Sensitivity of $R_I$, $R_p$, and DoLP (columns) to $r_{\mathrm{eff}}^{\mathrm{T}}$ in the VIS–NIR (rows) for pure snow (solid lines) and for snow containing impurities in the top layer with $C_{\mathrm{BC}}^{\mathrm{T}} = 5$ ppmw (dashed lines). The remaining parameters are $D^{\mathrm{T}} = 0.35$, $\rho^{\mathrm{T}} = 0.1\,\mathrm{g\,cm^{-3}}$, $Z^{\mathrm{T}} = 3$ cm, $r_{\mathrm{eff}}^{\mathrm{B}} = 320\,\mu\mathrm{m}$, $\mathrm{AR}^{\mathrm{B}} = 1.0$, $D^{\mathrm{B}} = 0.35$, $\rho^{\mathrm{B}} = 0.3\,\mathrm{g\,cm^{-3}}$, and $C_{\mathrm{BC}}^{\mathrm{B}} = 0$ ppmw. Calculations are for the principal plane and a SZA of 65°.

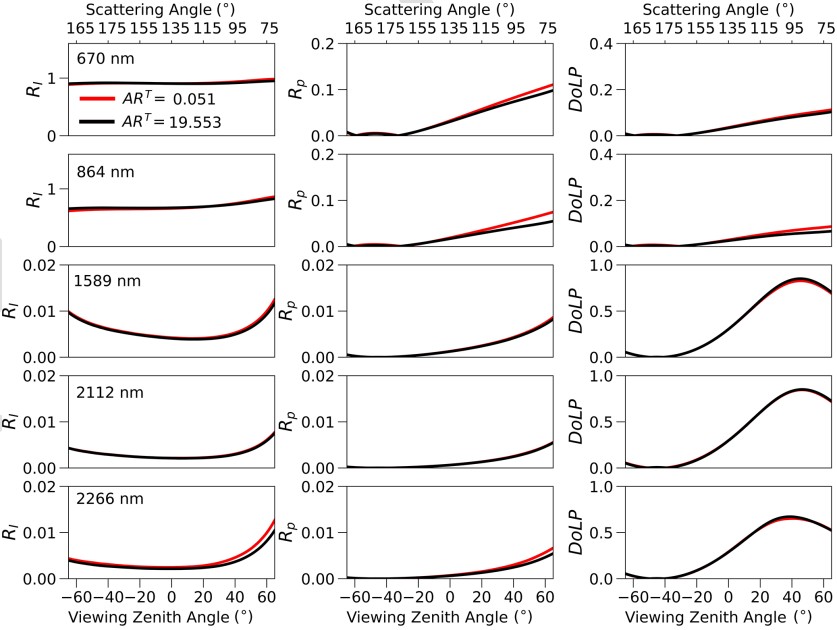

**Figure A3.** Comparison of $R_I$, $R_p$, and DoLP (columns) for a pure snowpack consisting of column crystals with $\mathrm{AR}^{\mathrm{T}} = 19.553$ (red) and plate crystals with $\mathrm{AR}^{\mathrm{T}} = 1/19.553 = 0.051$ (blue) at different wavelengths (rows). The remaining parameters are fixed at $r_{\mathrm{eff}}^{\mathrm{T}} = 1280\,\mu\mathrm{m}$, $D^{\mathrm{T}} = D^{\mathrm{B}} = 0.35$, $\rho^{\mathrm{T}} = 0.26\,\mathrm{g\,cm^{-3}}$, $Z^{\mathrm{T}} = 3$ cm, $r_{\mathrm{eff}}^{\mathrm{B}} = 320\,\mu\mathrm{m}$, $\mathrm{AR}^{\mathrm{B}} = 1.0$, and $\rho^{\mathrm{B}} = 0.4\,\mathrm{g\,cm^{-3}}$. Again, calculations are for the principal plane and a SZA of 65°.

*Code availability.* The radiative transfer code and the code used to perform the GSA, based on the SALib Python package (https://doi.org/10.5281/zenodo.160164, Usher et al., 2016), can be obtained from the corresponding author upon reasonable request.

*Data availability.* The complete look-up table of the forward simulations is stored on a dedicated NASA server at GISS in the form of a Python xarray object and can be obtained upon request.

*Author contributions.* MO led the study. GHM conceived the adaptation of the GSA formalism to the snow dataset and carried out all calculations, assisted by NC. MO wrote the main draft of the paper; all authors contributed to the editing.

*Competing interests.* The contact author has declared that none of the authors has any competing interests.

ther geographical representation in this paper. While Copernicus Publications makes every effort to include appropriate place names, the final responsibility lies with the authors.

*Acknowledgements.* Support from the NASA Office of STEM Engagement at the Goddard Institute for Space Studies, the Minority University Research and Education Project (MUREP) and the NASA internship program, is gratefully acknowledged CE2. The extra funding provided to Gabriel Harris Myers by Lucia Tsaoussi, program manager of the NASA Remote Sensing Theory program, was indispensable to finalize the study. All authors wish to thank Shenglong Wang and Valerio Luccio at the New York University for their patience and guidance in configuring the supercomputing architecture, Igor Geogdzhayev for the management of the GO database of the ice crystal optical properties, and Aaditya Rangan and Knut Stamnes for helpful discussions.

*Financial support.* This research has been funded through the NASA ROSES Remote Sensing Theory program (grant no. 80NSSC21K0569). Gabriel Harris Myers was supported by the NASA internship program and additional funding provided by Lucia Tsaoussi.

*Review statement.* This paper was edited by Alexander Kokhanovsky and reviewed by two anonymous referees.

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

**Remarks from the language copy-editor**

CE1    Please confirm the changes to the last two sentences.

CE2    Please confirm the changes to this sentence.

**Remarks from the typesetter**

TS1    Please give an explanation of why this needs to be changed. We have to ask the handling editor for approval. Thanks.

TS2    Please give an explanation of why this needs to be changed. We have to ask the handling editor for approval. Thanks.