# Peer review of "Global Sensitivity Analysis of simulated polarimetric remote sensing observations over snow"

_EGUsphere, 2023_

## Author Response (AR1)

First of all, we would like to sincerely apologize for the delay in delivering this response. It took a while to re-run the analyses on our large look-up table, which was necessary to address part of the reviewers' comments. We have also been impacted by the activities linked to the launch of the NASA/PACE satellite and illness at the time of the planned submission date for the response.

We thank the reviewers because their comments granted us the chance to double check all our computations, and we believe that this second draft is greatly improved. Note that the edits listed below required a number of other adjustments in the manuscript text, which are tracked in the new draft but are excluded from this response, since they do not affect the results or the methodology but only the flow.

We start from some general notes unrelated to the comments posed by the reviewers:

- The order of authors has been changed to reflect more faithfully each author's contribution up to this round of reviews.
- Throughout the entire manuscript, we have converted every instance of "light-absorbing impurity" (and the acronym LAI) to "light-absorbing particulate" (LAP). "Light-absorbing impurity" commonly refers to absorbing particulate embedded in the snow, but was used in the original manuscript to refer also to aerosols in the atmosphere above, causing unnecessary confusion.
- We have substituted everywhere (including in the plots) the symbol $\rho_C$ with $C_{BC}$, to avoid confusion between concentration of black carbon and snow density.
- We have added the missing VIS-NIR uncertainties in $R_p$ to the plots of the Sobol indices (Figs. 4, 7, 8, A1), and expanded the discussion in Sec. 2.2 (*lines 210-230*).

**Reviewer #1:**

**1) where is the proof that the forward simulations are correct? Before speaking about the retrieval of grain sizes and other parameters it would be nice to see a comparison of the radiative properties of some common scene measured and modeled with the help of the approach mentioned in the article**

The description of the forward simulations is already contained in the draft. The radiative transfer code is based on the Doubling-Adding formalism, and has been used to model Research Scanning Polarimeter (RSP) measurements over a variety of Earth's scenes for ~25 years, including scenes containing ice crystals in clouds [van Diedenhoven et al., 2013] and snow [Ottaviani et al., 2012; 2015]. The inherent optical properties of the ice crystals are calculated with an advanced Geometric Optics code

*(lines 123-127)*, as documented in many papers [Macke et al., 1996; van Diedenhoven et al., 2012].

Aerosols are treated as lognormal distributions of spherical particles, and so are the impurities in the snow that are also assumed to be externally mixed with the (hexagonal) snow grains *(lines 149-151)*. Within these assumptions, already listed in the paper, the code is exact to any arbitrary accuracy. We have added in the text that the code is plane parallel.

Ottaviani et al. [2012; 2015] demonstrated the feasibility of retrievals of snow grain shape, and microscale roughness based on RSP observations *(lines 56-60)*. In these studies, we isolated the surface contribution to the total signal measured at sensor altitude via an iterative procedure that automatically includes a rigorous atmospheric correction. By fitting the surface signal to the database of hexagonal prisms it was established that, radiatively speaking, snow behaves as a collection of non-spherical crystals with extreme aspect ratios. The reason for choosing hexagonal prisms is described in detail by the papers of van Diedenhoven *(lines 53-56)*.

The retrieval of grain size exploits measurements at infrared wavelengths, following the same strategy as the MODIS [Stamnes et al., 2007; Painter et al., 2009] or AVIRIS [Painter et al., 2003; Nolin and Dozier, 1993] teams. Such types of retrievals have been validated [Aoki et al., 2007; Painter et al., 2003].

We have made the following changes to the (new) Sec. 2.1:

*The plane-parallel RT code employed to generate the LUT is based on the general doubling-adding formalism described by De Haan et al. (1987). It features a consistent treatment of the radiative effects deriving from atmospheric molecular scattering, aerosols and clouds, and any surface whose reflectance is known in analytical form or in terms of its Bidirectional Reflectance Distribution Function (BRDF) properties and the polarization counterpart (BPDF). The code has been used for decades to model measurements from the RSP over a variety of Earth scenes, including those containing ice crystals in clouds (van Diedenhoven et al., 2013) and ground snow (Ottaviani et al., 2012; 2015).*

*…during multi-spectral retrievals (Li et al., 2001). In contrast to retrievals of grain size for mono-layer snowpacks (Nolin and Dozier, 1993; Painter et al., 2003), such an approach has been exploited to retrieve grain size in both a thin surface layer and a thick layer below using measurements from MODIS (Aoki et al., 2007; Painter et al., 2009).*

And to Sec. 3.1:

*The addition of NIR measurements of $R_l$ and of $R_p$ (if the accuracy of the latter matches RSP levels) gives access to information on grain shape and microscale roughness, as confirmed with real data (Ottaviani et al., 2012; 2015).*

**2) Do the retrieval methods mentioned in the manuscript provide a unique solution? I mean, can one get the same measurement for two different scenes (within instrumental noise level)? If the answer is positive then the authors should add the uncertainty linked to non-uniqueness of the solution to the methodological part.**

This is a general problem in remote sensing. Uniqueness problems are always to be expected in systems of elevated complexity. However, we note that optimal estimation methods are designed to provide the optimal solution in a statistical sense. The rigorous application of inverse methods also requires the user to provide the measurement uncertainties at input [Rodgers, 2000], in order to obtain the uncertainties on the retrieved parameters. Note that we considered measurement uncertainties of different sensors *(lines 33, 215-221, 313-319, 389, 411-413)*. For the test scenes, the error bars requested by the reviewer are exactly what is reported in Figs. 5, 6, 10 and 11.

In general, the inversions work as "optimally" as one is able to provide a good first guess, i.e. starting the inversion from a point reasonably close to the final solution, so as to avoid convergence to a local rather than the global minimum of the cost function. For the retrievals presented in the draft, the initial guess for each parameter was randomly selected between its upper and lower bounds. We have run some tests on the sensitivity to the initial guesses. As one example, we provide below the analog to Fig. 10, but with initial guesses closer to the true values: it can be seen that the inversion converges to the same results.

[Figure]

We made the following changes in the manuscript to address these arguments. In Sec. 3.1:

*Figure 5 summarizes the values of the state parameters and their uncertainty obtained from the inversion. The solid lines represent the "true" values used in the forward simulations. The dashed lines are instead the initial guess for each parameter, randomly sampled within the bounds listed in Table 1. The retrievals were repeated a few times to test the stability of the results against different initial guesses.*

and:
*…the larger uncertainty assigned to the simulated spaceborne measurements limits the retrieval quality compared to the RSP-like case. Finally, we note that all these retrievals are robust against different choices of the initial guess for each parameter.*

In Sec. 3.2:
*…confirming that polarimetric measurements in the SWIR are valuable for determining the vertical partitioning of LAPs. Initializing the inversion with first guesses close to the true values helps decreasing the uncertainty on $C_{BC}{}^T$ and $C_{BC}{}^B$ retrieved from the VIS+NIR+SWIR combination of $R_I$+$R_p$, but using $R_I$+DoLP still performs best.*

**(ii) Regarding the method, I would either add subsections to this part to distinguish the method from the scene description or, better yet, add a separate section dedicated to the object of study, including the instrument and the scene.**

We welcome this suggestion and have modified the Methods section so that it now contains the two subsections: 2.1 Radiative Transfer Simulations; and 2.2 Global Sensitivity Analysis Formalism.

**(iii) I would better formulate who is the end user of the information presented in the manuscript and how he/she will benefit from the information summarized in it. I believe, a paragraph or a section is needed, which would give clear instructions to the end user. In the present version of the manuscript, the conclusion contains just general phrases like "adding this band can improve the retrieval of that characteristic", whereas one expects to see more precise recommendations with actual numbers. Providing Sobol indices is important, but summarizing table and/or section would be more informative, I'd say. These recommendations should be highlighted both in the conclusion and in the abstract to make the work more useful to the community.**

A few sentences (*lines 37-39, 389-394, 406-407, 416-418*) mentioned that the augmented retrievals benefit climate modeling efforts. We added more explicit mention of end users.

In the abstract:

*The better characterization of surface and atmospheric parameters in the snow-covered regions advances the research opportunities for scientists of the cryosphere, and ultimately benefits the albedo estimates in climate models.*

And in the Conclusions:

*[...] the methods and results outlined in this paper provide cryospheric scientists with guidelines for selecting appropriate viewing geometries during data collection and for the development of advanced retrieval algorithms applied to airborne and spaceborne data over snow.*

To provide more specific guidelines, we have added "heatmaps" of the Sobol indices' to Sec. 3:

[Figure]

In these heatmaps, the cells are color-coded according to the maximum of the absolute Sobol index for each parameter (rows) and wavelength (columns) found in Figs. 7 and 8. The number indicates the corresponding viewing zenith angle. The top and bottom panels are for SZA=65° and SZA=45°. These figures can aid in the choice of appropriate viewing geometries and channel combinations when designing retrieval algorithms and observational strategies. Consequently, the text in Sec. 3.2 has been modified to:

*In the VIS-NIR, $\sigma_T^{RI}$ exhibits an essentially flat behavior well above the detection thresholds at all viewing zenith angles for many of the parameters, with shallow maxima at around nadir except for $AR^T$ and $\tau_c^{555}$. In the SWIR, $\sigma_T^{RI}$ and $\sigma_T^{RP}$ for $r_{eff}^T$ and $\tau_c^{555}$ peak*

*at the largest viewing zenith angles. The DoLP includes now sensitivity to $D^I$, occurring still in the forward-scattering half-plane but with peaks at smaller angles. Multi-angle polarization measurements can therefore greatly supplement those of total reflectance, especially when retrieving parameters that express marked angular differences in $\sigma_T$.*

*Figure 9 provides an alternative display of the information contained in Figs. 7 and 8, for SZA=65° (top panel) and SZA=45° (bottom panel). These heatmaps can aid in the choice of appropriate viewing geometries and channel combinations when designing retrieval algorithms and observational strategies. The intensity of each cell's color is proportional to the maximum value of $\sigma_T^{R_I}$ (left columns), $\sigma_T^{R_P}$ (middle columns), and $\sigma_T^{DoLP}$ (right columns) across all VZAs, and the number reports the angular location of these maxima. Numbers close to zero represent nadir-looking directions, and large positive angles correspond to the forward-scattering directions (see top x-axis in Figs. 7 and 8). It is evident how measurements in the forward-scattering half-plane are sensitive to the properties of aerosols and the top snow layer, while nadir-looking geometries favor the determination of parameters deeper in the snowpack. It is also clear how the addition of accurate polarimetric measurements in the VIS-SWIR benefits the retrieval of aerosol and surface properties, especially if at multiple angles.*

The Abstract has been modified to read:

*...in turn of the asymmetry parameter which is critical for the determination of albedo. The retrieval uncertainties are minimized when the Degree of Linear Polarization is used in place of the polarized reflectance.*

and the Conclusions to read:

*…can improve the characterization of processes like aerosol deposition in climate models and, again, albedo simulations. The findings generally promote the use of the DoLP over the polarized reflectance, and indicate that observations from the VIS all the way to the SWIR minimize the uncertainties when attempting to distinguish impurities in snow from absorbing aerosols.*

**(iv) Certain figures, namely Fig. 1, 4, 6, 10 leave a feeling that this space could have been used in a more informative way. For example, there are only 3 points on each panel of Fig. 6. Do we really need to build a plot in this case? Wouldn't it be more informative to present this information in a table or just in a text line? The information content of other figures is also small, compared to the place they take. I would try to pick up the most essential panels and present them in a simple way.**

Fig. 1: The external labels have been moved inside the layers to make the figure slimmer and decrease its overall size. All labels have been reformatted.

Figs. 4, 7, 8: The plots of the Sobol indices should be given in a consistent format, so that toggling among them aids their intercomparison. We have tried several options to produce such figures, and we actually think that the current version contains considerable information and is visually appealing, without being excessively cluttered. Note that the panels are arranged so that the legend fits without overlapping the curves or making the entire figure larger. Removing the empty panels in Fig. 4 defeats the purpose of highlighting the complex system of correlations stemming from the inclusion of impurities (Figs. 7 and 8). Note that we have added the missing uncertainties also in the VIS panels that do not show sensitivity to any of the parameters.

Figs. 5, 6, 10, 11: We have restricted the space between the vertical bars in Figs. 6 and 11 to make the figures more compact, as the reviewer suggests. The visual appearance of where the retrievals land (given the initial guesses) and the relative comparison of the error bars obtained using different instrument configurations is just simpler to perceive graphically, rather than having the reader parsing and comparing numbers in a table.

**Reviewer #2:**

**Line 34. The data used in the mentioned article is not SPEXone but POLDER/PARASOL.**

Thank you for catching this! It was a mistake coming from an earlier version of the draft. We corrected the relevant sentence to:

*Zhang et al. (2023) have recently evaluated the performance of introducing a snow kernel in an inverse algorithm to retrieve the microphysics of aerosols above snow based on observations of the Polarization and Directionality of the Earth's Reflectances (POLDER) spaceborne sensor, that flew from 2004 to 2013.*

**Table 1. The range for AOD which is used to generate the synthetic data is not appropriate. For aerosol over snow, the range is larger. It can be predicted that the overall accuracy would be quite similar because the distribution of AOD over snow can be regarded as log-normal in the global scale, but to make the synthetic scene more realistic, the case with larger AOD should also be assessed.**

The range of AOD was chosen to target amounts difficult to detect via remote sensing, and prevalent in climatologically significant regions like Greenland. This strategy was mentioned during the retrieval tests, but not clearly enough elsewhere. For consistency, we also restricted the range of snow impurity concentration in the LUT to a maximum of

1 ppmw. To address the reviewer's comment, we then performed the GSA also on a LUT generated with an extended range of AOD (up to 1.2) and impurity density (up to 10 ppmw), and attempted retrievals simulated for $\tau_C^{555} = 1.0$ and $C_{BC}^T = C_{BC}^B = 2$ ppmw. The results are nearly identical to those presented in Figs. 7, 10, and 11, and the sensitivity to $\tau_c^{555}$ at polarimetric measurements in the SWIR is even more pronounced.

[Figure]

The text added to Sec. 3.2 reflects these changes:

*…their accurate determination is especially important for climate modeling (Antwerpen et al., 2022; Wang et al., 2020; Alexander et al., 2014; Ryan et al., 2019). To target these challenging retrievals, in Fig. 7 the results of the GSA are computed for maximum LAP loads of 0.4 for $\tau_c^{555}$ and 1 ppmw for $C_{BC}^T$ and $C_{BC}^B$, where the subscript "BC" indicates the specific type of black-carbon LAP considered in this paper, with fixed microphysical and optical properties. More sporadic events like thick burning plumes or exceptionally dirty snow are addressed in the Appendix, where the same calculations are repeated with extended ranges of $\tau_c^{555}$ (up to 1.2) and $C_{BC}^T$ and $C_{BC}^B$ (up to 10 ppmw). At these higher LAP amounts, the sensitivity of polarimetric measurements in the SWIR to aerosol optical depth is even more pronounced (see Fig. A1 and related discussion).*

And to the appendix (Fig. A1):

[Figure]

*This Appendix discusses a few aspects that could not be included directly in the main text of the paper without unnecessarily interrupting the flow. Figure A1 is the same as Fig. 7, but considers a larger range of aerosol optical depth (up to 1.2) and impurity density (up to 10 ppmw), which can occur as a result of burning biomass plumes or extremely "dirty" snow. With higher concentrations of LAPs, the absolute Sobol indices for the visible and NIR wavelengths increase (see Eq. 17) because BC absorption causes large variations in the total reflectance and, to a more limited extent, in the polarized reflectance especially at larger viewing zenith angles (see also Fig. 3 in Ottaviani (2022)). Larger DoLP signals also lead to a minor increase in the uncertainty threshold, but the list of parameters with identified sensitivity in Fig. 7 remains the same.*

*…partly correlated to the dimension of the crystals. The differences in the Sobol indices compared to Fig. 7 are only minor. Polarimetric measurements in the SWIR show selective sensitivity to $\tau_c^{555}$, $r_{eff}^T$, and $D^T$, and the sensitivity to $\tau_c^{555}$ is even more prominent. The general conclusions drawn in the main text remain therefore unaffected if a larger range of LAPs is considered.*

**For section 2, the authors consider the snow impurity but ignore the problems caused by mixed pixels consist of other landcover type besides snow. This situation should be at least mentioned in the discussion part.**

The primary goals of this paper are (i) to showcase the GSA as a suitable tool for the analysis of the information content of complex, hyperdimensional datasets; and (ii) discuss the application to snow cases, which is largely unexplored from the point of view of modern remote sensing. We have chosen to include the differences between highly idealized and more realistic scenarios by considering cases with and without LAPs. The explicit target of optically semi-infinite snow is mentioned in the introduction (*lines 90-91, caption of Fig. 1*), and the discussion of heterogeneous pixels is deemed out of scope. We have followed the reviewer's recommendation by adding to Sec. 3.1:

*In both cases, we consider optically semi-infinite snowpacks since the focus of this paper is on the remote sensing of snow; heterogeneous pixels constitute an added layer of complexity and will be the subject of future studies.*

and:

*As explained in Sec. 2, the snowpack consists of a mixture of crystals ($f^T$=$f^B$=0.5). Fresher snow (smaller grains) is simulated in the top layer ($r_{eff}^T$=150 $\mu m$, $Z^T$=3 cm, $\rho^T$=0.2 g/cm$^3$, $AR^T$=0.05 for plates and corresponding $1/AR^T$=20 for columns, $D^T$=0.3 (Ottaviani, 2015)). More compact, larger and rounder grains are located in the bottom layer ($r_{eff}^B$=250 $\mu m$, $\rho^B$=0.30 g/cm$^3$, $AR^B$=0.15 for plates and 6.67 for columns, $D^B$=0.40), which is optically semi-infinite ($\tau \approx 2000$).*

And to the Conclusions:

*The GSA presented in this study can be extended to LUTs that consider a whole suite of aerosol optical properties, region-specific impurity amounts and more elaborate mixing schemes (Tanikawa et al. 2019), or optically thin snowpacks with different underlying land cover types.*

**To validate your theory, I believe that an experiment with real data is important, and the polarimetric data over snow is available in the community. I believe that after validation your algorithm would be more convincing.**

We first redirect this concern to the first response given to the other reviewer. The RT code has already been shown to successfully model RSP measurements over snow. The retrievals simulated in this paper are consistent with plausible polar scenarios.

Also, the results of the GSA are independent of validation datasets, and serve as a theoretical background for experiments with real data. We are currently in the process of merging the only extensive spaceborne polarimetric dataset available (POLDER) to MODIS observations, but much more work is required to set up, batch-process and validate the retrievals. Such effort is beyond the scope of this paper.